# Effects of Seed Crystals on the Growth and Catalytic Performance of TS-1 Zeolite Membranes

**DOI:** 10.3390/membranes10030041

**Published:** 2020-03-13

**Authors:** Wenjuan Ding, Sitong Xiang, Fei Ye, Tian Gui, Yuqin Li, Fei Zhang, Na Hu, Meihua Zhu, Xiangshu Chen

**Affiliations:** State-Province Joint Engineering Laboratory of Zeolite Membrane Materials, Institute of Advanced Materials, College of Chemistry and Chemical Engineering, Jiangxi Normal University, Nanchang 330022, China; 1607640062@jxnu.edu.cn (W.D.); 2513782178@jxnu.edu.cn (S.X.); 1185992702@jxnu.edu.cn (F.Y.); 12617482@jxnu.edu.cn (T.G.); 1033424625@jxnu.edu.cn (Y.L.); 270828010@jxnu.edu.cn (F.Z.); 84815294@jxnu.edu.cn (N.H.)

**Keywords:** TS-1 zeolite membrane, seed crystals, catalytic oxidation, mono-layer

## Abstract

Dense and good catalytic performance TS-1 zeolite membranes were rapidly prepared on porous mullite support by secondary hydrothermal synthesis. The properties of seed crystals were very important for the preparation of high-catalytic performance TS-1 zeolite membranes. Influences of seed crystals (Ti/Si ratios, size, morphology, and zeolites concentration of the seed suspension) on the growth and catalytic property of TS-1 zeolite membranes were investigated in details. High Ti/Si ratio, medium-size, and morphology of the seed crystals were critical for preparing the high-performance TS-1 zeolite membrane. Compared with the bi-layer TS-1 zeolite membrane (inner and outer of the mullite tube), the mono-layer TS-1 zeolite membrane had a better catalytic performance for Isopropanol IPA oxidation with H_2_O_2_. When the Ti/Si ratio, size, and morphology of the TS-1 zeolites were 0.030, 300 nm, ellipsoid, and the zeolites concentration of the seed suspension was 5%, the IPA conversion, and flux through the TS-1 zeolite membrane were 98.23% and 2.58 kg·m^−2^·h^−1^, respectively.

## 1. Introduction

Taramasso et al. prepared the titanium silicalite-1 (TS-1) with MFI structure by the hydrothermal synthesis in 1983, which was a breakthrough in the zeolites field [1]. TS-1 zeolites could catalyze the liquid-phase oxidation of a variety of organic compounds to oxygenated products actively, which has been focused on the TS-1/H_2_O_2_ reaction system for the environmentally benign and outstanding oxidation selectivity [2,3,4,5,6,7].

Separation and recovery of the TS-1 zeolites from the liquid-phase reactions are a major problem, which is limited to the industrial application of TS-1 zeolites. TS-1 zeolite membrane could avoid the separation and recovery processes [8,9,10], which is a novel and good catalyst for H_2_O_2_ oxidations with some advantages, such as mild conditions (low reaction temperature and room pressure), environment-friendly, and energy-saving [11,12]. 

Compact and good catalytic performance TS-1 membranes have been successfully prepared by in-situ or secondary growth hydrothermal synthesis [13,14,15]. Secondary growth hydrothermal synthesis, a layer of seeded zeolites is coated on suitable support before crystallization, which is an effective way for shortening the crystallization time and preparing high-quality zeolite membranes [16,17,18,19,20]. In addition, the secondary growth could suppress the transformation from nucleation to other phase crystals. It is well known that the seeded crystals play an important role in the membrane formation. *b*-oriented TS-1 zeolite membranes were prepared by the secondary growth method, which had a high reproducibility and excellent catalytic performance for the n-hexane oxyfunctionalization [16]. Zhu et al. first prepared the Ti-MWW zeolite membrane by secondary hydrothermal synthesis, and the membranes showed good catalytic performance for phenol hydroxylation [17]. Liu et al. synthesized the highly oriented, thin, and hierarchically porous TS-1 zeolite membrane by secondary hydrothermal treatment [18]. Wang et al. synthesized the Ti-containing membrane using nanosized silicalite-1 particles as seeds by hydrothermal treatment; the membranes showed large proportions of tetrahedrally coordinated titanium and excellent hydrothermally stability [19].

Recently, the high catalytic performance TS-1 zeolite membrane with the ultra-dilute synthetic solution by the secondary hydrothermal synthesis in our previous study, which greatly shortened the crystallization time and preparation cost of the membrane [20]. In order to improve the catalytic performance of the membranes, effects of the seed crystals (Ti/Si ratios, size, morphology, and zeolites concentration of the seed suspension) on the growth and catalytic performance of the TS-1 zeolite membranes are studied in the present work. Besides, influences of the bi-layer/mono-layer TS-1 zeolite layer on the catalytic performance of the TS-1 zeolite membrane were investigated in this work.

## 2. Experimental Section

### 2.1. Materials

The silicon sources, titanium sources, and organic structure-directing agent (SDA) were tetraethyl orthosilicate (Sigma-Aldrich, TEOS, MO, USA, 98 wt%), tetra-tert-butyl orthotitanate (TCI, TTIP, Tokyo, Japan, 97 wt%), and tetrapropylammonium hydroxide (TCI, Tokyo, Japan, TPAOH, 10 wt% in H_2_O). The porous mullite tubes (Noritake Co., Nagoya, Japan, OD: 12 mm, ID: 9 mm, average pore size: 1.3 μm, length: 100 mm) were used as supports. Six types of MFI zeolite crystals were used as seeded crystals for preparing TS-1 zeolite membranes, which were numbered Z-1 (ellipsoid, ca. 230 nm, Ti/Si = 0); Z-2 (ellipsoid, ca. 300 nm, Ti/Si = 0.022); Z-3 (ellipsoid, ca. 300 nm, Ti/Si = 0.030); Z-4 (agglomerate, ca. 540 nm, Ti/Si = 0.030); Z-5 (petals, ca. 540 nm, Ti/Si = 0.030); Z-6 (reunion, ca. 750 nm, Ti/Si = 0.030), respectively.

### 2.2. Preparation of TS-1 Zeolite Membrane 

The seed crystals were coated on the mullite support by dip-coating. Seed crystals were dispersed into ethanol and formed 2–7 wt% zeolite crystals suspension, and the solution was ultrasonicated to a uniform solution. The mullite supports were seeded with TS-1 zeolite crystals suspensions before hydrothermal synthesis. The support was immersed in the zeolite crystals suspension for 40 s by two times, the seeded support was placed and dried at 85 °C ovens. It was noted that the pipe mouth of the mullite support was blocked during the seeding procedure, which could prevent the zeolite crystals to enter into the inner surface of the support. The molar composition of the precursor synthesis solution was SiO_2_: 0.035 TiO_2_: 0.25 TPAOH: 120 H_2_O and the preparation procedure of the precursor synthesis solution was identical with our previous work [20]. The seeded supports were still plugged with Teflon rods during crystallization procedures in this study. Thereafter, the precursor synthesis solution and seeded supports were sealed in a titanium autoclave at 423 K for 24 h. The membranes were washed, dried and calcined after hydrothermal synthesis. The calcination was carried out at 500 °C for 10 h; the heating and cooling rate were 0.25 °C/min. 

### 2.3. Characterizations

X-ray diffraction (XRD, Rigaku Ultima IV, Tokyo, Japan) with Cu-Kα radiation, Fourier transformed infrared spectroscopy (FT-IR, JASCO FT/IR-6100, Tokyo, Japan), ultraviolet, visible spectroscopy (UV-vis, JASCO, V-700, Tokyo, Japan) and field-emission scanning electron microscopy (FE-SEM, Tokyo, Japan) were characterized the structure and morphology of TS-1 zeolites and membranes. Energy Dispersive X-Ray Spectroscopy (EDX, Bruker Quantax200, Bruker, Germany) and inductively coupled plasma atomic emission spectroscopy (ICP-AES, Varian 725ES, California, USA) were measured the element composition of zeolites.

Oxidation of isopropanol (IPA) and H_2_O_2_ was carried in a 500 mL membrane reactor with a reflux apparatus, and the reaction device, as illustrated in our previous study [20]. Firstly, 5.27 g IPA, 96.34 g H_2_O_2_ (30%) and 254.39 g H_2_O were added into the membrane reactor; the TS-1 zeolite membrane was introduced into the membrane reactor. The water bath was heated to 70 °C by the water bath. The reaction equation was as follows.



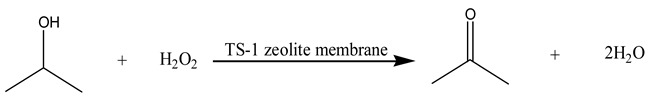



The total flux (*Q*) and IPA conversion (*C_IPA_*) were calculated as the following:*Q* = *W*/(*A* × *t*)(1)
*C_IPA_* = *n_acetone_*/(*n_IPA_* + *n_acetone_*) × 100%(2)
where *W*, *t*, and *A* were the total weight of the permeate (kg), collecting interval (h), and effective surface area of TS-1 zeolite membrane (m^2^), *n_acetone_* and *n_IPA_* were the molar contents of acetone and IPA in the permeate mixture, respectively. It was noted that the acetone selectivity of IPA oxidation by the TS-1 zeolite membrane was almost 100% in this work. The oxidation product was periodically collected and analyzed by gas chromatography (GC, Shimadzu, GC-2014C, Tokyo, Japan).

## 3. Results and Discussion

Six types of TS-1/Silicalite-1 zeolites were applied as seed crystals (Z-1, Z-2, Z-3, Z-4, Z-5 and Z-6) for preparing TS-1 zeolite membranes. Table 1 summarized the properties of the seed crystals (morphology, size, and Ti/Si), and Figure 1 presented SEM images of these zeolites. Zeolite Z-1 was the typical silicalite-1, which was ellipsoid with an average size of ca. 230 nm. Zeolites Z-2 and Z-3 exhibited well-defined edges and regular ellipsoid morphology (ca. 300 nm), while the Ti/Si ratios of the zeolites Z-2 and Z-3 were 0.022 and 0.030 by ICP characterization. Besides, in order to study effects of morphology on the growth of TS-1 zeolite membrane, zeolites Z-4 (ca. 540 nm, agglomerate), Z-5 (ca. 540 nm, petals) Z-6 (ca. 750 nm, reunion) were used as seed crystals, whose Ti/Si ratio were the same as zeolites Z-3. 

### 3.1. Effect of Seed Crystals Ti/Si Ratio

Table 2 summarized the preparation conditions and catalytic performance of TS-1 zeolite membranes in this work. Water was the only by-product of isopropanol/hydrogen peroxide by TS-1 zeolite membranes, and all the membranes showed good perm-selectivity for the acetone by pervaporation (PV) nearly 100%. Figure 2 and Figure 3 showed the XRD patterns, surface, and cross-section SEM images of the TS-1 zeolite membranes with different Ti/Si ratio seed crystals.

As given in Figure 2, all diffraction peaks of the membranes M-1, M-2, M-3 were the MFI zeolite structure (2θ = 7.8°, 8.8°, 23.2°, 23.8°) and mullite support diffraction peaks, which indicated that all membranes had pure MFI phase [21,22,23,24]. Besides, Figure 3 presented that the mullite surface was fully covered with the TS-1 zeolites. 

Table 2 summarized the Ti/Si ratio of the membranes by EDX characterization, which was increased with the Ti/Si ratio of the seeded crystals. The Ti/Si ratio of the membrane M-1 was low (0.024, Table 2), the catalytic performance of the membrane M-1 was poor, the total flux and IPA conversion of membrane M-1 were only 1.79 kg·m^−2^·h^−1^ and 37.78%. The accompanying TS-1 zeolites of the TS-1 zeolite membranes were collected in this work, the number of the accompanying zeolites of the membranes M-2 and M-3 were S-2 and S-3. Figure 4 showed the FT-IR spectra of the TS-1 zeolites. The peak at ca. 960 cm^−1^ of the spectra was ascribed to the interaction between the stretching vibration of [SiO_4_] unit and titanium in neighboring coordination sites, which was an evidence of the vibration of Si–O–Ti bond in the zeolite framework. As given in Figure 4, the zeolite S-3 had a higher tetrahedral titanium adsorption peak (ca. 960 cm^−1^) than the zeolite S-2. Besides, the M-3 had a higher Ti/Si ratio than the M-2 by EDX characterization (Table 2), the total flux of the membrane M-3 and IPA conversion of oxidation were up to 2.58 kg·m^−2^·h^−1^ and 98.18%, respectively. Hence, the high Ti/Si ratio seed crystals were a favor for preparing fine crystals layer and good catalytic performance TS-1 zeolite membrane in this work. Table 3 presented the amounts of seeded crystals, seeded support, and membrane M-3. ca. 0.62 g seeded crystals were attached to the support by dip-coating. After crystallization, washing, drying, and calcination, the weight of the zeolite layer was ca. 0.75 g. 

### 3.2. Effect of Seed Crystals Morphology

Morphology and size of the seeded crystals had a great influence on the growth and performance of the zeolite membrane by secondary hydrothermal synthesis [19,25]. The TS-1 zeolite membranes (M-3, M-4, M-5, and M-6, Table 2) were prepared with different morphology and size TS-1 zeolites (Z-3, Z-4, Z-5, and Z-6, Table 1). Figure 5 and Figure 6 were the XRD patterns, surface and cross-section SEM images of the membranes M-4, M-5, and M-6.

XRD patterns of theses TS-1 zeolite membranes (M-4, M-5, and M-6) indicated that the characteristic diffraction peaks belonged to the MFI-type zeolites diffraction peaks. IPA conversion of the membranes was high by the medium size and morphology TS-1 zeolites membrane (M-3, 98.18%, M-5, 86.70%); the fine seed crystals was conducive to prepare good catalytic performance TS-1 zeolite membrane in this work. The aggregated zeolites Z-4 and Z-6 had a large size; therefore, the mullite support was difficult to adsorb large aggregated seeded crystals from the suspension. Hence, there were many large size zeolites on the membranes, and the surface of the membranes (M-4 and M-6) were relatively rough (Figure 6a,b,e,f). The particles on the surface of the membrane were large, the contact area with the reactant molecules became small, and the catalytic performance of the membranes was poor. 

### 3.3. Effect of Seed Crystals Suspension Concentration 

The mullite support was seeded with TS-1 zeolites by dip-coating; the concentration of seeded crystals suspension was critical for preparing dense TS-1 zeolite membrane [26]. Figure 7 presented the surface and cross-sectional images of the seeded supports with different seeds concentration suspension. All mullite support surface was fully covered with the fine and ellipsoid Z-3 zeolites, while the thickness of the seed crystals layer was increased with the concentration of the seeded crystals in the suspension. As shown in Table 2, the membranes M-7, M-8, M-3, and M-9 were prepared with different concentration seed suspension, and the concentration of seeded crystals suspension was 2%, 3%, 5%, and 7%. When the concentration of seeded crystals suspension was 5%, the membrane M-3 had the highest catalytic performance in this work. Figure 8 displayed the XRD patterns of these TS-1 zeolite membranes, the diffraction patterns indicated that the random orientation of the TS-1 zeolite membrane was formed on the porous mullite support, and intensity of the MFI structural diffraction peaks was increased with the seed crystals suspension concentration.

As shown in Figure 3e,f, and Figure 9, the TS-1 zeolite crystals layer thickness and morphology of the membranes (M-7, M-8, M-3, and M-9) were dependent on the seeded crystals suspension concentration. When the concentration of seeded crystals suspension was 5%, the zeolite layer thickness of the membrane M-3 was ca. 18 μm, and the membrane M-3 had the highest catalytic performance for the much catalytic active sites [27] because the support surface had a certain adsorption limitation and saturation of zeolite crystals. Even though the seeded crystals suspension concentration was reached 7%, excessive zeolite crystals of the support surface were dissolved into the precursor gel, and the large zeolites and intercrystalline pores were formed on the membrane M-9 surface, which had poor performance for the IPA oxidization (Table 2). 

### 3.4. Comparison of Bi-Layer and Mono-Layer TS-1 Zeolite Membranes

In our previous study, the bi-layer TS-1 zeolite membrane had a good catalytic performance for IPA and H_2_O_2_ [20]. As shown in Figure 10a,b, the out and inner support surface were fully covered by the dense TS-1 zeolite layer. Mass transfer resistance was depended on the thickness of the membrane layer, the preparation of the mono-layer membrane was a favor for increasing the membrane flux. Because the nano zeolite crystals could easily enter the mullite support (average pore size, ca. 1.3 μm) and grow into a bi-layer TS-1 zeolite membrane, which could increase the mass transfer resistance of the membrane. In order to prepare a thin TS-1 zeolite membrane layer, the mullite tubes were added with the PTFE rods during the crystallization process in this work. As shown in Figure 10c,d, a dense and fine TS-1 zeolite layer was formed on the outer surface of the support; there were only a few seed crystals were scattered on the inner surface of the support, which could not form a continuous zeolite layer. Table 2 exhibited the catalytic performance of the bi-layer (M-10) and mono-layer (M-3) TS-1 zeolite membranes, the flux and IPA conversion of the membrane M-10 was only 1.98 kg·m^−2^·h^−1^ and 93.27%. The membrane M-3 had the plenty of fine TS-1 zeolite crystals and a thin mono-layer zeolite layer, the flux and IPA conversion of were reached 2.58 kg·m^−2^·h^−1^ and 98.23%. 

Table 4 summarized the catalytic performance of five pieces pf mono-layer TS-1 zeolite membrane with the zeolite Z-3. Clearly, the membranes had a similar catalytic performance, which showed the mono-layer TS-1 membrane preparation had good reproducibility in this work. Besides, titanosilicate stability was important for the TS-1 zeolite membrane [28]. The repeatable catalytic performance of the mono-layer and bi-layer membranes (M-3 and M-10) was studied in this work (Table 5). The IPA conversion of the two membranes showed no obvious changes, indicating that the catalysts were robust catalysts because the structure destruction, loss of framework Ti, or the coke deposition of the catalyst [29,30], both the flux of the membranes were decreased with the running times, and the declining of the bi-layer TS-1 membrane flux (M-10) was larger than the mono-layer TS-1 membrane (M-3). The mono-layer TS-1 zeolite membrane presented the better catalytic performance and repeatability than the bi-layer zeolite membrane, particularly the flux. 

## 4. Conclusions

In summary, a facile and green approach to produce acetone by a mono-layer TS-1 zeolite membrane catalyst had been developed. The effects of seed crystals on the growth and properties of TS-1 zeolite membranes were investigated. When the zeolite seed crystal had a suitable Ti/Si ratio, size, morphology and seed suspension concentration (0.030, 300 nm, ellipsoid, 5%, respectively), a dense and mono-layer TS-1 zeolite membrane was fully covered on the seeded mullite support, the IPA conversion and the flux of TS-1 zeolite membrane were 98.23% and 2.58 kg·m^−2^·h^−1^. 

## Figures and Tables

**Figure 1 membranes-10-00041-f001:**
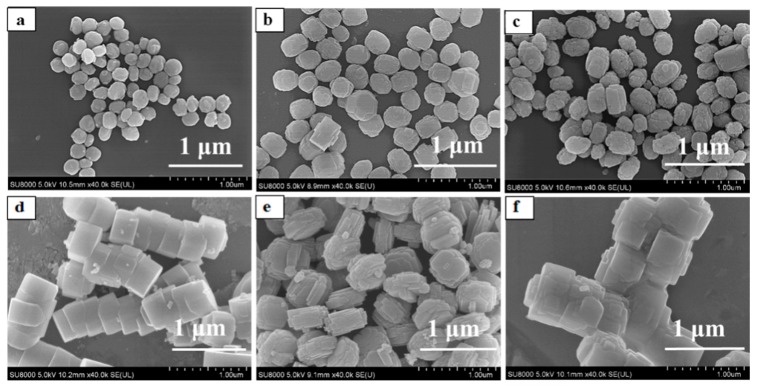
SEM images of Silicalite-1/TS-1 zeolites. (**a**) Z-1; (**b**) Z-2; (**c**) Z-3; (**d**) Z-4; (**e**) Z-5; (**f**) Z-6.

**Figure 2 membranes-10-00041-f002:**
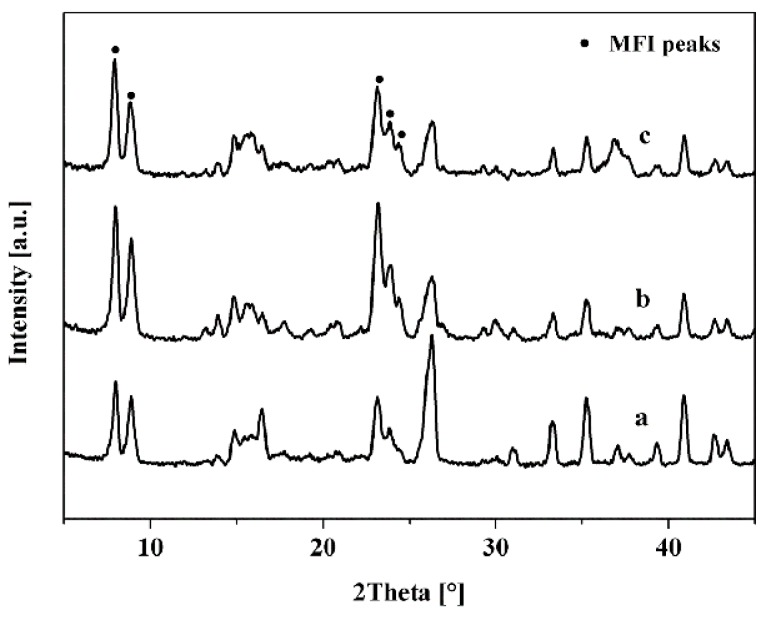
XRD patterns of TS-1 zeolite membranes with different Ti/Si ratio seeded crystals. (**a**) M-1; (**b**) M-2; (**c**) M-3.

**Figure 3 membranes-10-00041-f003:**
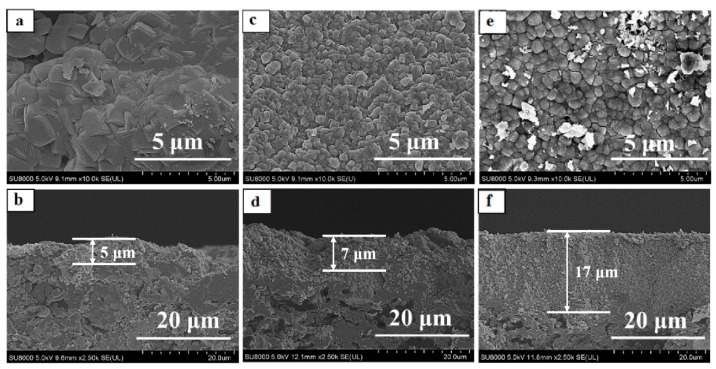
Surface and cross-section SEM images for TS-1 zeolite membranes with different Ti/Si ratio seed crystals, (**a**,**b**) M-1; (**c**,**d**) M-2; (**e**,**f**) M-3.

**Figure 4 membranes-10-00041-f004:**
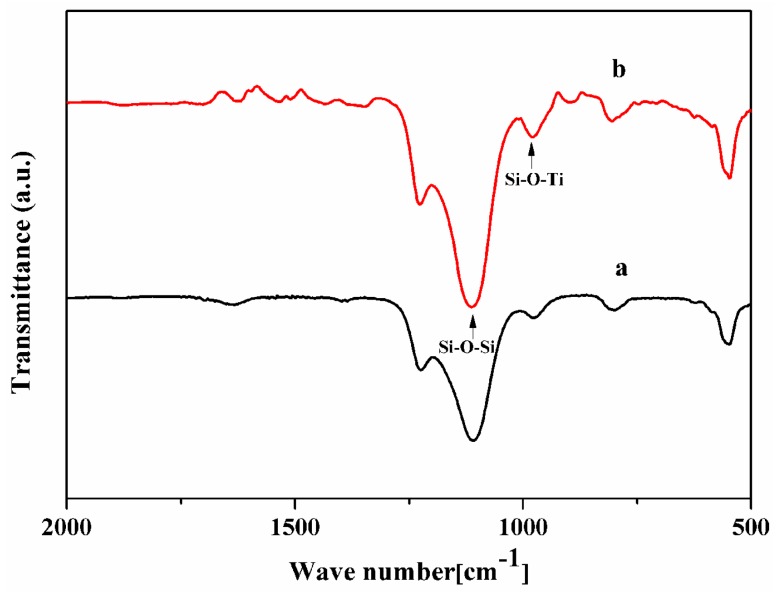
FT-IR spectra of the accompanying TS-1 zeolites: (**a**) S-2; (**b**) S-3.

**Figure 5 membranes-10-00041-f005:**
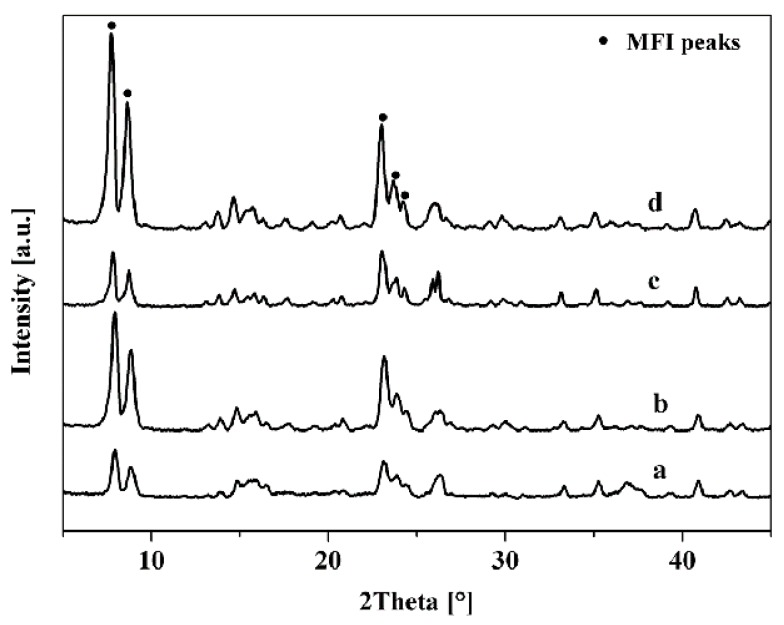
XRD patterns of TS-1 zeolite membranes with different morphology of seeded crystals. (**a**) M-3; (**b**) M-4; (**c**) M-5; (**d**) M-6.

**Figure 6 membranes-10-00041-f006:**
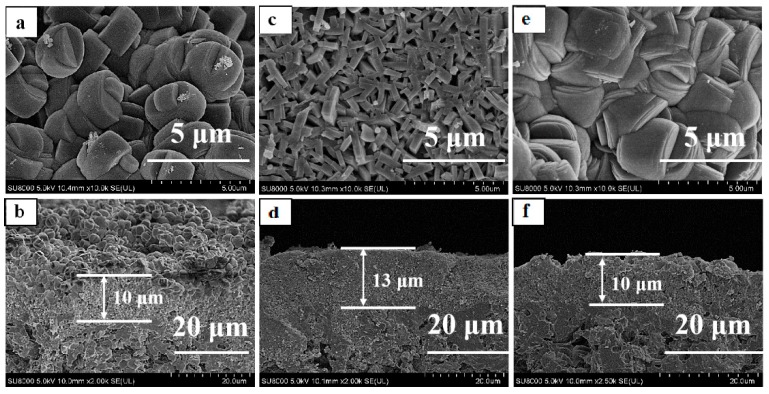
Surface and cross-section SEM images for TS-1 zeolite membranes with different morphology of seeded crystals. (**a**,**b**) M-4; (**c**,**d**) M-5; (**e**,**f**) M-6.

**Figure 7 membranes-10-00041-f007:**
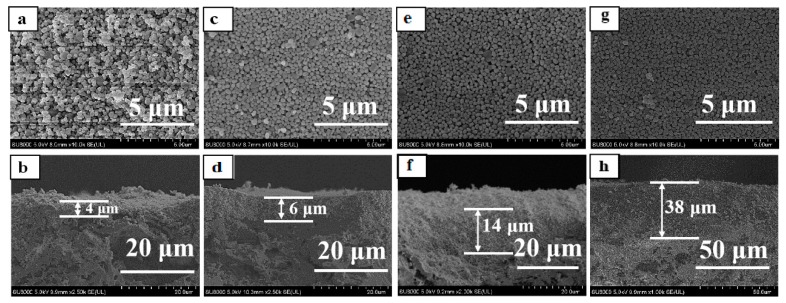
Surface and cross sectional images of seeded support with different seeded crystals concentration suspension (Z-3). (**a**,**b**) 2%; (**c**,**d**) 3%; (**e**,**f**) 5%; (**g**,**h**) 7%.

**Figure 8 membranes-10-00041-f008:**
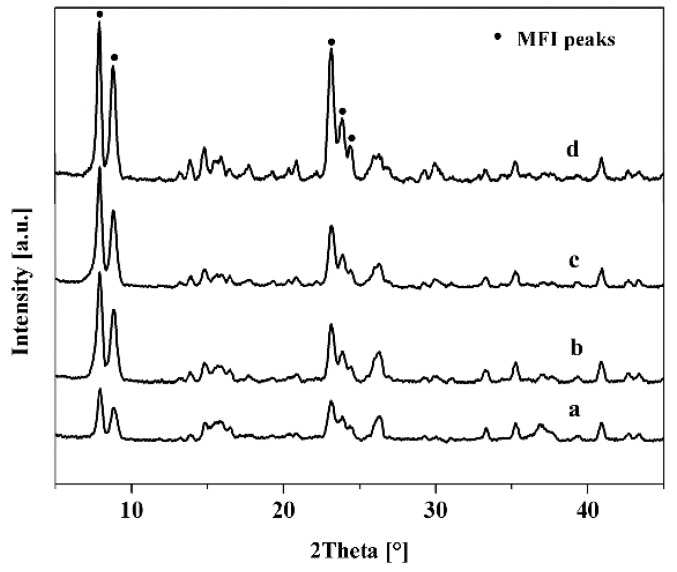
XRD patterns of TS-1 zeolite membranes with with different seeded crystals concentration suspension. (**a**) 2%, M-7; (**b**) 3%, M-8; (**c**) 5 catalytic %, M-3; (**d**) 7%, M-9.

**Figure 9 membranes-10-00041-f009:**
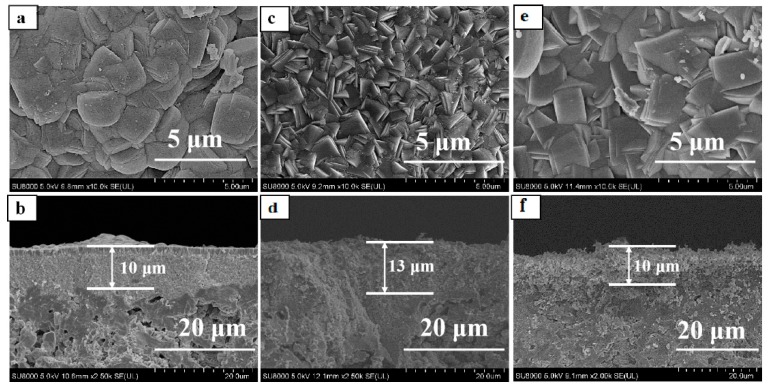
Surface and cross-section SEM images for the TS-1 membranes prepared with different seeded crystal suspension concentrations. (**a**,**b**) 2%, M-7; (**c**,**d**) 3%, M-8; (**e**,**f**) 7%, M-9.

**Figure 10 membranes-10-00041-f010:**
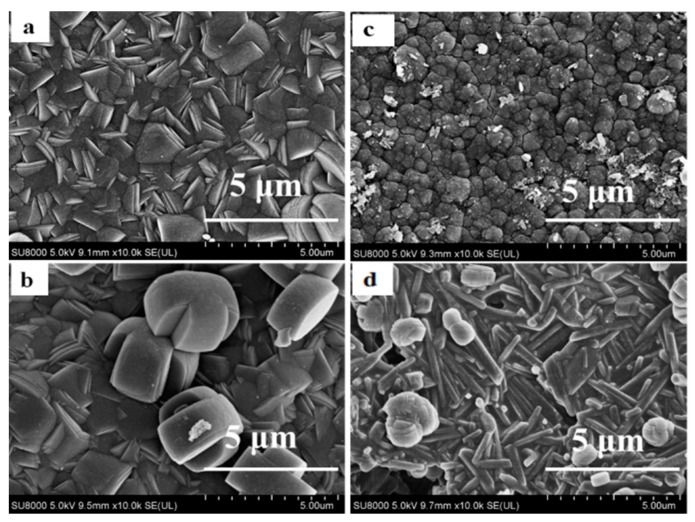
Outer (**a**,**c**) and inner (**b**,**d**) surface images of the bi-layer (M-10, a and b) and mono-layer (M-3, c and d) TS-1 zeolite membranes.

**Table 1 membranes-10-00041-t001:** Textural properties of Silicalite-1/TS-1 zeolites.

No.	Morphology	Size ^2^ (nm)	Ti/Si ^1^ Ratio
Z-1	ellipsoid	230	0
Z-2	ellipsoid	300	0.022
Z-3	ellipsoid	300	0.030
Z-4	agglomerate	540	0.030
Z-5	petals	540	0.030
Z-6	reunion	750	0.030

Note: ^1^ determined by Inductively coupled plasma atomic emission spectroscopy (ICP), ^2^ size of a single crystal block.

**Table 2 membranes-10-00041-t002:** Catalytic performance and Ti/Si ratio of the TS-1 zeolite membranes with different seeded crystals.

Membrane No.	Seed Suspension		Ti/Si Ratio	Q [kg·m^−2^·h^−1^]	C_IPA_ [%]
	Seed No.	Seed Concentration (wt%)			
M-1	Z-1	5%	0.024	1.79	37.78
M-2	Z-2	5%	0.036	1.90	81.88
M-3	Z-3	5%	0.055	2.58	98.18
M-4	Z-4	5%	0.027	1.84	69.02
M-5	Z-5	5%	0.0319	1.97	86.70
M-6	Z-6	5%	0.024	2.80	64.89
M-7	Z-3	2%	0.025	1.06	68.72
M-8	Z-3	3%	0.035	1.29	82.56
M-9	Z-3	7%	0.030	Leak	Leak
M-10 *	Z-3	5%	0.051	1.98	93.27

Note: * M-10 is the bi-layer TS-1 zeolite membrane.

**Table 3 membranes-10-00041-t003:** Amount of the support, seeded support, and final membrane M-3.

Support (g)	Seeded Support (g)	Amount of Seed (g)	Membrane (g)	Amount of Zeolite (g)
9.44	10.06	0.62	10.19	0.75

**Table 4 membranes-10-00041-t004:** Catalytic performance of TS-1 zeolite membranes in IPA/H_2_O_2_ mixture at 70 °C.

No.	Q (kg·m^−2^·h^−1^)	C (%)
M-3	2.58	98.18
M-11	2.63	97.34
M-12	2.30	93.87
M-13	2.48	96.64
M-14	2.18	97.29

Note: The molar composition of the synthesis gel of the TS-1 zeolite membrane was SiO_2_: 0.035 TiO_2_: 0.25 TPAOH: 120 H_2_O, the crystallization temperature and time were 150 °C and 1 d.

**Table 5 membranes-10-00041-t005:** Repeatable catalytic performance of TS-1 zeolite membranes (M-3 and M-10).

Running Times	M-3		M-10	
	Q (kg·m^−2^·h^−1^)	C (%)	Q (kg·m^−2^·h^−1^)	C (%)
1	2.58	98.23	1.98	93.27
2	2.45	98.18	1.82	93.04
3	2.34	97.91	1.62	93.65

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
