# Peer review of "Effects of Seed Crystals on the Growth and Catalytic Performance of TS-1 Zeolite Membranes"

_membranes, 2020, doi:10.3390/membranes10030041_

Round 1

Reviewer 1 Report

In this manuscript, authors demonstrated the preparation of TS-1 membrane on a mullite support with using silicalite-1/TS-1 seed crystals. The effect of seed properties has discussed, however, it seems the discussion was not scientific and lacking experimental evidences. Following concerns should be eliminated before publication.

Major comment 1. In line 158-160. Authors discussed the incorporation of Ti by newly appeared XRD peak at 29.2°. The discussion is skeptical and not reasonable. How did you assigned the peak as the Ti in the zeolite framework. I also checked the reference 24, but no such a discussion or insight was given.

Major comment 2. In the preparation of the membrane, the deposition of seed was conducted by immersing the mullite support in the seed crystal suspension. The method is unclear and seems to be difficult to reproduce the result. Did you washed the support after picking it from the suspension? Did you dried the sample before the second immersing? Is it different from a conventional "dip coating" approach? 

Major comment 3. Related to the comment 1, the amount of seed deposited on the support should be shown in the manuscript if possible. 

Major comment 4. In the lines 200-205, the low catalytic performance of M-4 and M-6 was explained by the size of crystal particle formed on the surface and resulting small contact area with the reactant molecule. The discussion here is a little jumpy. Was this a discussion about "surface roughness" of the membrane? or something else? Because this is a "membrane" and reactants go thorough the layer, why did size of the particle affect the performance?  

And, what does "compactness (in line 195)" and "compact (in line 201)" means? I could not catch the meaning of "compact layer".

Author Response

Respond to the reviewers comments

Reviewer #1:

Dear Reviewer,

Thank you for your letter and for the reviewer’ comments concerning our manuscript entitled “Effects of seed crystals on the growth and catalytic performance of TS-1 zeolite membranes” (ID:membranes-733183). Those comments are valuable and very helpful for revising and improving our paper, as well as the important guide to our researches. We studied those comments carefully and made corrections of the manuscript. We hope that the revision could meet with your approval. The main corrections in the manuscript was marked in Blue in the paper, and the responds to your comments are as following:

In this manuscript, authors demonstrated the preparation of TS-1 membrane on a mullite support with using silicalite-1/TS-1 seed crystals. The effect of seed properties has discussed, however, it seems the discussion was not scientific and lacking experimental evidences. Following concerns should be eliminated before publication.

Major comment 1. In line 158-160. Authors discussed the incorporation of Ti by newly appeared XRD peak at 29.2°. The discussion is skeptical and not reasonable. How did you assigned the peak as the Ti in the zeolite framework. I also checked the reference 24, but no such a discussion or insight was given.

Reply: Thank you for pointing out this.The article has been adjusted accordingly.

“The unit cell parameters of TS-1 were found to increase linearly as a function of titanium content for experimental data having been obtained by least-squares fit to interplanar spacings of selected reflections in the X-ray diffraction pattern. The monoclinic symmetry of the crystals was easily detected by observing the splitting of some reflections in the XRD pattern (for example, the reflection located at 2θ = 24.4 ° and 29.3°).” ( J. Catal. 1992,137(2), 497-503.)

Major comment 2. In the preparation of the membrane, the deposition of seed was conducted by immersing the mullite support in the seed crystal suspension. The method is unclear and seems to be difficult to reproduce the result. Did you washed the support after picking it from the suspension? Did you dried the sample before the second immersing? Is it different from a conventional "dip coating" approach? 

Reply:Thank you for pointing out this. The details of the seeded procedure of the support was added in the manuscript (Section 2.2). The support was immersed in the zeolite crystals suspension for 40 s for two times, and then the seeded support was placed and dried at 85 â„ƒ oven. It is noted that the pipe mouth of the mullite support was blocked during seeding procedure, which could prevent the zeolite crystals enter into the inner surface of the support. The support was not washed after from the suspension, and the sample was dried before the second immersing, which was similar with the conventional “dip coating” approach.

Major comment 3. Related to the comment 1, the amount of seed deposited on the support should be shown in the manuscript if possible. 

Reply:Special thanks for your comments. The amount of seed deposited on the support was added in the manuscript. Table 4 summarized the seed amount of the support, seed support and membrane M-3. The seed crystals of 0.52g were attached to the support by dip-coating. After crystallization, washing, drying and calcination, the  zeolite layer weight was 0.75g, which guaranteed that TS-1 zeolite membrane had a high catalytic performance. (Section 3.4)

Major comment 4. In the lines 200-205, the low catalytic performance of M-4 and M-6 was explained by the size of crystal particle formed on the surface and resulting small contact area with the reactant molecule. The discussion here is a little jumpy. Was this a discussion about "surface roughness" of the membrane? or something else? Because this is a "membrane" and reactants go thorough the layer, why did size of the particle affect the performance?  

Reply:Special thanks for your comments. As shown in Figure S1, the zeolites Z4 and Z6 contain much large aggregated crystals, the mullite support is difficult to adsorb large aggregated seeds from the seed suspension. Hence, there are much large size zeolites on the membranes, and surface of the membranes (M-4 and M-6) are relatively rough. For the final morphology of the film, the nucleation stage is an important factor, size distribution of the zeolite crystals have a major influence[2].  Besides, the catalytic oxidation reaction of TS-1 is restricted by the internal diffusion of zeolites, the small crystal had the high surface utilization and catalytic oxidation activity[3]. Besides, the M-4 and M-6 membrane have low Ti/Si ratios by EDX characterization (Table 2). Therefore, the M-4 and M-6 films have poor catalytic performance.

Fig. S1 SEM images of TS-1 zeolites. (a) Z-4; (b) Z-6.

And, what does "compactness (in line 195)" and "compact (in line 201)" means? I could not catch the meaning of "compact layer".

Reply:Special thanks for your comments. "Compactness (in line 195)" and "Compact (in line 201)" means that there are dense zeoite layer on the support surface. As shown in Figures 3e and 3f, Figures 6c and 6d, the zeolite crystals are good inter-growth.  

References

  1. Millini, R.; Massara, E.P.; Perego, G.; Bellussi, G.Framework composition of titanium-silicalite-1.J. Catal. 1992,137(2), 497–503. https://doi:10.1016/0021-9517(92)90176-i.
  2. Wendler, F.;Mennerich, C.; Nestler, B. A phase-field model for polycrystalline thin film growth.  Cryst. Growth. 2011, 327(1), 18189–201. https://doi:10.1016/j.jcrysgro.2011.04.044.
  3. Van der Pol, A. J. H. P.;Verduyn, A. J.; van Hooff, J. H. C. Why are some titanium silicalite-1 samples active and others not? Appl. Catal. A. 1992, 92(2), 113–130. https://doi:10.1016/0926-860x(92)80310-9. 

We tried our best to improve the manuscript and made some changes in the manuscript.  These changes will not influence the content and framework of the paper. We appreciate for your warm work earnestly, and hope that the correction will meet with approval. Once again, thank you very much for your comments and suggestions.

Thank you and best regards.

Yours sincerely,

Mei-Hua Zhu

Reviewer 2 Report

The paper “Effects of seed crystals on the growth and catalytic performance of TS-1 zeolite membranes” is a study on TS-1 zeolites prepared on porous mullite support by secondary hydrothermal synthesis. Authors investigated the influences of several parameters of seed crystals (Ti/Si ratios, size, morphology, concentration of suspension) on the growth and on the catalytic activity for IPA oxidation with H2O2, of TS-1 membranes. Membranes were characterized by FESEM, UV-vis, EDX and XRD.  In addition, they compared bi-layer and mono-layer TS-1 zeolite. The paper gives some information that may be of interest in the synthesis of good TS-1 catalysts, but it needs a deep revision of the English language, in addition to minor and major revisions.

1)           English need a deep revision. There are errors in syntax (especially), grammar, plurals/singular, conjugation of verbs, punctuation.

2)           Experimental: Seed crystals, are they purchased or synthetized by authors? there are not clear information about that. Paper need more information on these crystals. Availability or preparation methods.

3)           The experimental need some more information about the parameters of the chromatographic analysis of the oxidation products.

4)           Table 1. Z-3 reported with wrong Ti/Si.

5)           The Ti/Si column should be filled for every material, for clarity.

6)           Authors should give more details about the analysis used for measuring the Ti/Si composition reported in the table1: EDX or ICP?

7)           Z-5 seeds are defined as petals morphology (fig. 1e). In my opinion, It seems not true. It seems a bigger ellipsoidal morphology. Authors should modify or justify with reference this attribution to petals.

8)           Authors write (284-287)  “When the zeolite seed crystal has a suitable Ti/Si ratio, size, morphology and seed suspension concentration, a dense and mono-layer TS-1 zeolite membrane was fully covered on the seeded mullite support, the IPA conversion and the flux of TS-1 zeolite membrane were 98.23% and 2.58 kg·m-2·h -1.”

The conclusions are clearly not exhaustive, authors must enlarge and improve the conclusion paragraph, reporting which are these suitable parameters. 

Author Response

Respond to the reviewers comments

Reviewer #2:

Dear Reviewer,

Thank you for your letter and for the reviewer’ comments concerning our manuscript entitled “Effects of seed crystals on the growth and catalytic performance of TS-1 zeolite membranes” (ID:membranes-733183). Those comments are valuable and very helpful for revising and improving our paper, as well as the important guide to our researches. We studied those comments carefully and made corrections of the manuscript. We hope that the revision could meet with your approval. The main corrections in the manuscript was marked in Blue in the paper, and the responds to your comments are as following:

The paper “Effects of seed crystals on the growth and catalytic performance of TS-1 zeolite membranes” is a study on TS-1 zeolites prepared on porous mullite support by secondary hydrothermal synthesis. Authors investigated the influences of several parameters of seed crystals (Ti/Si ratios, size, morphology, concentration of suspension) on the growth and on the catalytic activity for IPA oxidation with H2O2, of TS-1 membranes. Membranes were characterized by FESEM, UV-vis, EDX and XRD.  In addition, they compared bi-layer and mono-layer TS-1 zeolite. The paper gives some information that may be of interest in the synthesis of good TS-1 catalysts, but it needs a deep revision of the English language, in addition to minor and major revisions.

  • English need a deep revision. There are errors in syntax (especially), grammar, plurals/singular, conjugation of verbs, punctuation.

Reply: We have revised the whole manuscript carefully and tried to avoid the grammar or tense error. In addition, we have asked several colleagues who are skilled authors of English language to check the paper. Special thanks to you for your good comments.

  • Experimental: Seed crystals, are they purchased or synthetized by authors? there are not clear information about that. Paper need more information on these crystals. Availability or preparation methods.

Reply: Thank you for pointing out this. Table S1 summarizes the properties, source and molar composition of the precursor synthesis gel of the seed TS-1 zeolites crystals. Besides, the seed crystals were dispersed into ethanol and formed 2~7 wt% zeolite crystals suspension, the mullite supports were seeded with TS-1 zeolite crystals before hydrothermal synthesis.

Table S1 Properties, source and molar composition of the precursor synthesis gel of the seed TS-1 zeolites crystals.

No.

Morphology

Size (nm)

Ti/Si ratio    

Seed source

Molar composition of the synthesis gel

Z-1

ellipsoid

230

0

Home-made

SO2:0.36TPAOH:20H2O

Z-2

ellipsoid

300

0.022

Home-made

SO2:0.03TiO2:0.35TPAOH:28H2O

Z-3

ellipsoid

300

0.030

Shanghai Gechi chemical Co. LTD

---

Z-4

agglomerate

540

Home-made

SO2:0.01TiO2:0.35TPAOH:28H2O

Z-5

petals

540

Home-made

SO2:0.035TiO2:0.35TPAOH:28H2O

Z-6

reunion

750

Home-made

SO2:0.04TiO2:0.35TPAOH:28H2O

  • The experimental need some more information about the parameters of the chromatographic analysis of the oxidation products.

Reply:Thank you for pointing out this. When TS-1 zeolite membrane was used in the process of coupled pervaporation-catalytic oxidation reaction, the permeation liquid was characterized by GC-MS technology in this work.The test results showed that most of our products were acetone.

  • Table 1. Z-3 reported with wrong Ti/Si.

Reply:The Z-3 zeolite was purchased from Shanghai gechi co., LTD and Ti-Si ratios obtained by the ICP, which were performed on a Varian 725-ES.

  • The Ti/Si column should be filled for every material, for clarity.

Reply: Thank you for pointing out this.The article has been adjusted accordingly.

  • Authors should give more details about the analysis used for measuring the Ti/Si composition reported in the table1: EDX or ICP?

Reply:Table 1 shows the different Ti/Si ratios obtained by elemental analysis of the zeolite samples using ICP measurements, which  were performed on a Varian 725-ES. Table 2 summarizes the Ti/Si ratios of the membrane by EDX.

  • Z-5 seeds are defined as petals morphology (fig. 1e). In my opinion, It seems not true. It seems a bigger ellipsoidal morphology. Authors should modify or justify with reference this attribution to petals.

Reply: Thank you for pointing out this. As shown in Figure 1, the morphology of Z-5 has a similar ellipsoidal shape, but the surface was extremely rough, which may be composed of agglomeration of smaller particles, the morphology of petals was observed [1].

8) Authors write (284-287)  “When the zeolite seed crystal has a suitable Ti/Si ratio, size, morphology and seed suspension concentration, a dense and mono-layer TS-1 zeolite membrane was fully covered on the seeded mullite support, the IPA conversion and the flux of TS-1 zeolite membrane were 98.23% and 2.58 kg·m-2·h -1.” The conclusions are clearly not exhaustive, authors must enlarge and improve the conclusion paragraph, reporting which are these suitable parameters. 

Reply: Thank you for pointing out this.The article has been adjusted accordingly.

References

  1. Pang,C.;Xiong, J.; Li,G.; Hu, C. Direct ring C-H bond activation to produce cresols from toluene and hydrogen peroxide catalyzed by framework titanium in TS-1. J. Catal. 2018, 366, 37–49, https://doi:10.1016/j.jcat.2018.07.038.

We tried our best to improve the manuscript and made some changes in the manuscript.  These changes will not influence the content and framework of the paper. We appreciate for your warm work earnestly, and hope that the correction will meet with approval. Once again, thank you very much for your comments and suggestions.

Thank you and best regards.

Yours sincerely,

Mei-Hua Zhu

Round 2

Reviewer 1 Report

The manuscript was updated. I have added comments just for minor updates.

Comment 1,

About the discussion at XRD diffraction at 29.3°, authors answered as follows. “The unit cell parameters of TS-1 were found to increase linearly as a function of titanium content for experimental data having been obtained by least-squares fit to interplanar spacings of selected reflections in the X-ray diffraction pattern. The monoclinic symmetry of the crystals was easily detected by observing the splitting of some reflections in the XRD pattern (for example, the reflection located at 2θ = 24.4° and 29.3°).” (J. Catal. 1992,137(2), 497-503.)

I understand that. The point of my question is that the peak at 29.3° is a conventional (hkl)=(352) diffraction observed over any MFI type structure. Thus, it is impossible to say the "Ti-atom" is incorporated in the structure without careful refinement analysis. I recommend simply remove the related discussion from the manuscript, and Ti-incorporation does not need to be explained by such a skeptical discussion as far as Ti/Si ratio is listed on the table 2. And also it is recommended to remove "Ti peak" from Figure 2.

Comment 2.

In line 259, the seed-supporting method was described as "dip-coating". It would be good to use same expression in the experimental section and here. 

Comment 3.

The table 4 is difficult to follow. I recommend to use the terms

"Seed support" --> "Seeded support"

"Seed amount" --> "Amount of seed"

And I believe the "amount of seed" should be "0.62 g", isn't it?

"zeolite amount" --> "Amount of zeolite".

And also, you should state the sample is about M-X using Z-3 in the Table caption or somewhere in the table, not only in the main text.

Comment 4.

Related to above comment 3, it would be good to show the Table 4 in a very beginning of the manuscript. This is a structural feature of the membrane, and it would be good for readers to understand how the membrane look like.

And, the expression in line 260, "which guaranteed that TS-1 zeolite membrane had a high catalytic performance." is jumpy. I recommend to remove this part.

Author Response

Reviewer #1:

Dear Reviewer,

Thank you for your letter and for the reviewer’ comments concerning our manuscript entitled “Effects of seed crystals on the growth and catalytic performance of TS-1 zeolite membranes” (ID:membranes-733183). Those comments are valuable and very helpful for revising and improving our paper, as well as the important guide to our researches. We studied those comments carefully and made corrections of the manuscript. We hope that the revision could meet with your approval. The main corrections in the manuscript was marked in Red in the paper, and the responds to your comments are as following: 

Comment 1,

About the discussion at XRD diffraction at 29.3°, authors answered as follows. “The unit cell parameters of TS-1 were found to increase linearly as a function of titanium content for experimental data having been obtained by least-squares fit to interplanar spacings of selected reflections in the X-ray diffraction pattern. The monoclinic symmetry of the crystals was easily detected by observing the splitting of some reflections in the XRD pattern (for example, the reflection located at 2θ = 24.4° and 29.3°).” (J. Catal. 1992,137(2), 497-503.)

I understand that. The point of my question is that the peak at 29.3° is a conventional (hkl)=(352) diffraction observed over any MFI type structure. Thus, it is impossible to say the "Ti-atom" is incorporated in the structure without careful refinement analysis. I recommend simply remove the related discussion from the manuscript, and Ti-incorporation does not need to be explained by such a skeptical discussion as far as Ti/Si ratio is listed on the table 2. And also it is recommended to remove "Ti peak" from Figure 2.

 Reply: Thank you for pointing out this.The article has been adjusted accordingly, and have removed "Ti peak" from Figure 2,5,8. The accompanying TS-1 zeolites of the TS-1 zeolite membranes were collected in this work, the number of the accompanying zeolites of the membranes M-2 and M-3 were S-2 and S-3. Figure 4 showed the FT-IR spectra of the TS-1 zeolites. The peak at ca. 960 cm-1 of the spectra was ascribed to the interaction between the stretching vibration of [SiO4] unit and titanium in neighboring coordination sites, which was an evidence of the vibration of Si-O-Ti bond in the zeolite framework.

Comment 2.

In line 259, the seed-supporting method was described as "dip-coating". It would be good to use same expression in the experimental section and here. 

 Reply: Thank you for pointing out this.The article has been adjusted accordingly.

Comment 3.

The table 4 is difficult to follow. I recommend to use the terms

"Seed support" --> "Seeded support"

"Seed amount" --> "Amount of seed"

And I believe the "amount of seed" should be "0.62 g", isn't it?

"zeolite amount" --> "Amount of zeolite".

And also, you should state the sample is about M-X using Z-3 in the Table caption or somewhere in the table, not only in the main text.

  Reply: Thank you for pointing out this.The article has been adjusted accordingly.

Comment 4.

Related to above comment 3, it would be good to show the Table 4 in a very beginning of the manuscript. This is a structural feature of the membrane, and it would be good for readers to understand how the membrane look like.

And, the expression in line 260, "which guaranteed that TS-1 zeolite membrane had a high catalytic performance." is jumpy. I recommend to remove this part.

Reply: Thank you for pointing out this.The article has been adjusted accordingly, Table 4 are removed to the part.

We tried our best to improve the manuscript and made some changes in the manuscript.  These changes will not influence the content and framework of the paper. We appreciate for your warm work earnestly, and hope that the correction will meet with approval. Once again, thank you very much for your comments and suggestions.

Thank you and best regards.

Yours sincerely,

Mei-Hua Zhu

Reviewer 2 Report

The paper has been improved and it may be published in the present form.

Author Response

Dear Reviewer,

Thanks very much for your recognition on our work. Besides, thank you for your letter and for the reviewer’ comments concerning our manuscript entitled “Effects of seed crystals on the growth and catalytic performance of TS-1 zeolite membranes” (ID:membranes-733183). Those comments are valuable and very helpful for revising and improving our paper, as well as the important guide to our researches. 

We have revised the whole manuscript carefully and tried to avoid the grammar or tense error. In addition, we have asked several colleagues who are skilled authors of English language to check the paper. Special thanks to you for your good comments.

Once again, thank you very much for your comments and suggestions.

Thank you and best regards.

Yours sincerely,

Mei-Hua Zhu

This manuscript is a resubmission of an earlier submission. The following is a list of the peer review reports and author responses from that submission.

Round 1

Reviewer 1 Report

The manuscript by Ding et al. reports the influence of the TS-1 seeds on the performance of TS-1 membranes for the oxidation of isopropanol in the presence of hydrogen peroxide. Although many results are reported, there is no an attempt to explain the results and it is not clear whether these results are reproducible. For instance, Z-2 and Z-3 seeds have similar sizes and morphology and slightly different Ti/Si ratio, however, the Ti/Si ratio of the TS-1 membranes obtained using these seeds (Table 3) is quite different. The effect of the seeds on the performance is the focus of the manuscript, yet, no details of how the seeds have been prepared nor studies of the seeded substrates are provided. Z4 and Z6 seeds consist of aggregated crystals and it is not clear how this effects the dip coating procedure (for which details are missing as well). Without the above information, no general conclusions about the effect of the seeds on the performance of the membranes can actually be drawn. Therefore, I do not recommend this manuscript for publication.

Reviewer 2 Report

The manuscript conducted the detailed parametric study in the impact of seed crystal under the synthesis of TS-1 membrane supported on mullite. Several key parameters about seed crystals including Ti/Si ratio, size, morphology, and concentration in a suspension, were compared and the most appropriate condition was investigated. Although the study was conscientious, it seems like the comparison was not enough and the conclusion was not general. Before accepting this manuscript followings should be updated.

1) Only 6 seed samples (Table 1) compared in this study might not be enough to compare all the parameters tested in this manuscript. Otherwise, all the separated sections 3.1-3.4 can be summarized in one Table.

2) Please provide all the Ti/Si ratio in Table 1, which should be critical in the discussion in 3.1. And, you compared the catalytic performance based on the IPA conversion. I am wondering whether this is meaningful or not. Please provide and compare the reaction based on the reaction rate per Ti amount.

3) How much amount of zeolites were formed? Please provide weight % of initial seed amount compared to the final product.  

4) In the discussion in 3.2, I can not find "plenty amount of amorphous" from Figs 5a and 5e. And also, what is the definition of "medium size" and "medium morphology"? 

5) Please provide information about M-9 and M-10 in Table 2.

6) In all the cross-section SEM images, it would be helpful to indicate where is the zeolite layer. 

7) In line 228, what is "Figure 9c"?

8) I am wondering about the reproducibility of this membrane. Can you prepare the membrane with similar performance in several times?